# Influence of Microstructure and Alloy Composition on the Machinability of α/β Titanium Alloys

**DOI:** 10.3390/ma16020688

**Published:** 2023-01-10

**Authors:** Mostafa M. Shehata, Shimaa El-Hadad, Mahmoud Sherif, Khaled M. Ibrahim, Ahmed I. Z. Farahat, Helmi Attia

**Affiliations:** 1Central Metallurgical Research and Development Institute, Helwan P.O. Box 87, Egypt; 2National Research Council, Montreal, QC H4P 2R2, Canada; 3Department of Mechanical Engineering, McGill University, Montreal, QC H3A 0C3, Canada

**Keywords:** titanium alloys, WEDM, surface characteristics, modeling

## Abstract

A comparative study was conducted for the machining of two α/β titanium alloys, namely Ti-6Al-4V (Ti64) and Ti-6Al-7Nb (Ti67), using wire electric discharge machining (WEDM). The influence of cutting speed and cutting mode on the machined surfaces in terms of surface roughness (Ra), recast layer (RL), and micro-hardness have been evaluated. Rough cut (RC) mode at a cutting speed of 50 µm/s resulted in thermal damage; Ra was equal to 5.68 ± 0.44 and 4.52 ± 0.35 µm for Ti64 and Ti67, respectively. Trim-cut mode using seven cuts (TRC-VII) at the same speed decreased the Ra to 1.02 ± 0.20 µm for Ti64 and 0.92 ± 0.10 µm for Ti67. At 100 µm/s, Ra reduced from 2.34 ± 0.28 µm to 0.88 ± 0.12 µm (Ti64), and from 1.42 ± 0.15 µm to 0.90 ± 0.08µm (Ti67) upon changing from TRC-III to TRC-VII. Furthermore, a thick recast layer of 30 ± 0.93 µm for Ti64 and 14 ± 0.68 µm for Ti67 was produced using the rough mode, while TRC-III and TRC-VII modes produced layers of 12 ± 1.31 µm and 5 ± 0.72 µm for Ti64 and Ti67, respectively. Moreover, rough cut and trim cut modes of WEDM played a significant role in promoting the surface hardness of Ti64 and Ti67. By employing the Response Surface Methodology, it was found that the machining mode followed by cutting speed and the interaction between them are the most influential parameters on surface roughness. Finally, mathematical models correlating machining parameters to surface roughness were successfully developed. The results strongly promote the trim-cut mode of WEDM as a promising machining route for two-phase titanium alloys.

## 1. Introduction

Titanium and its alloys are the preferred materials for engineering and aeronautical applications. It is utilized to make turbine blades and other engine components, and it is also employed in the biomedical industry to make implant materials. Titanium alloys have good resistance to corrosion, a high strength/weight ratio, outstanding biocompatibility, and the ability to maintain quality at high temperatures [1]. Also, titanium has lower thermal conductivity, electrical resistance, and thermal expansion values when compared to other metals [2].

Machining Ti-alloys using conventional milling and turning has been challenging due to wear of the cutting tools and poor surface finish [3]. The deterioration of the Ti-alloy surface and subsurface after machining may have an obvious influence on the tribological properties and fatigue life of engineered components [4]. In the case of biomedical applications, surface properties, such as roughness and chemical composition, affect cell/tissue adhesion, and hence surface finish is highly considered for machined Ti-implants [5]. The chemical composition has an observed effect on the material’s thermal and mechanical properties and, thus, predictable practical performance [6]. However, typical machining techniques that are used to mill and remove materials by plastic shearing (for example, broaching, drilling, milling, turning), abrasion (for example, lapping, grinding), or micro-chipping (for example, polishing micro-abrasive, and blasting) from difficult-to-machine materials such as titanium-based alloys, in general, are expensive. Titanium alloys suffer from excessive tool wear during the cutting process, increased machining time, and poor machinability due to their inherent mechanical and thermo-physical properties [7]. Low thermal conductivity, high chemical reactivity, and low modulus of elasticity are the major drawbacks in machining titanium with conventional processes. Thus, a prominent non-conventional machining technique, like wire electrical discharge machining (WEDM), is utilized as a feasible and more economical alternative to generate complicated shapes from hard materials.

The basic principle of WEDM is that material is removed through rapid repetitive electrical discharges, which takes place between a continuously moving thin wire electrode and the workpiece. During WEDM, a pulse voltage difference is utilized between the working material and the electrode, both of which have to be immersed in a suitable dielectric fluid. At a certain voltage and a gap between the electrode and workpiece, a column of electromagnetic flux is formed, called a plasma channel, with energy densities around 10^11^–10^14^ W/m^2^. Due to this high electromagnetic energy density, very high temperatures have topically developed that range between 6000–12,000 °C, resulting in the melting and/or ablation of material from both the workpiece and the electrode [8,9]. The amount of metal removed during this spark discharge is determined by the desired cutting speed and surface quality. The heat from each electrical spark erodes a little part of the material that is vaporized and melted from the workpiece, as well as some of the wire material. A stream of de-ionized water is used to wash these particles (chips) away from the cut via the top and bottom flushing nozzles. The water also keeps heat from building up in the workpiece. Thermal expansion of the component might impact the size and positioning accuracy if not cooled [10]. Due to the rapid heating and cooling in WEDM, material re-solidification occurs (also known as recast layer), resulting in brittleness and cracks, micro-porosity, and residual stresses in the top layer of the machined surface [11]. Furthermore, because of the constant heating of the material, the heat-affected zone (HAZ) forms, which leads to grain coarsening, strength decrease, and a change in the microstructure of the machined surface [12].

WEDM has been reported as a successful machining process in which hard-to-cut materials are machined with high accuracy, especially in complex shapes. Moreover, it is a contactless machining process wherein no contact occurs between the working electrode and the workpiece. Thus, no cutting forces are developed, leading to machined workpiece material free of residual stresses. The clamping pressure required to hold tiny, thin, and fragile components is low, reducing workpiece damage or distortion, in addition to the capability to use computer-assisted numerical control to achieve three-dimensional shapes [13]. Due to the obvious advantages of WEDM compared to conventional cutting processes, it can manufacture any electrically conductive material, regardless of hardness, from tool steel, aluminum, copper, and graphite to exotic space-age alloys like Inconel, titanium, carbide, conductive ceramics, etc. Moreover, it is extensively used in the production of stamping and extrusion tools, dies, and molds in the automotive and aerospace industry, as well as in the production of medical components [14].

WEDM is a complex process with multi-parameters, including electrical parameters: peak current, pulse on time (T_on_), pulse off time (T_off_), supply voltage (V), and polarity, non-electrical parameters: speed of wire; rate of feed, machining duration, gain and flushing rate, parameters of electrode: material (copper, graphite, and tungsten) and wire size. Additionally, there are parameters related to the dielectric system: viscosity, type, and other characteristics. Figure 1 depicts the Ishikawa diagram of cause/effect, which shows the process factors that might impact the quality of machining, cutting, or drilling in the Wire EDM process [15].

It is well known that the performance of WEDM process is assessed principally by the machined Surface Quality (SQ), which can be significantly controlled using trim-cutting procedures in WEDM. It has been reported that there is an obvious difference in both micro-hardness and surface roughness of metallic alloys under the trim and rough-cut mode of WEDM [16]. After using a rough cut, the trim cut mode is applied to remove the thin surface layer that is thermally affected and consists of re-deposited material induced in rough cutting [17]. However, with several finish cuts, the average surface roughness and recast layer thickness could be significantly reduced [18]. In addition, the surface quality decreased with increasing cutting speed [19].

Researchers have recently determined that electrical discharge machining (EDM) and WEDM may be used to alter the surface topography of metallic biomaterials to enhance their bio-functionality as implants [20]. According to the researchers [21,22], EDM- modified surface improved cell attachment between the implant and the human tissues. In the in vitro study of Mahajan [23], it was observed that EDM surface surpassed the untreated surface in terms of performance and surface roughness. EDM process was found to soften the material up to a depth of nearly 200 µm under the white layer. WEDMed surfaces are distinguished by high hardness and low elastic modulus, which achieve the correlation between the wear resistance and the (Hardness/Modulus of elasticity) ratio [24]. According to Fonda et al. [25], the production rate in machining Ti-6A-4V (the most often used titanium alloy) using EDM is closely related to the workpiece temperature during machining, implying that appropriately determining machining parameters are critical.

In the context of using titanium alloys in biomedical applications, Ti-6Al-7Nb alloy (a two-phase alloy with Nb as a β-stabilizing element) has been recently standardized (ASTM standard F 1295) to avoid the reported cytotoxicity of V in Ti-6Al-4V alloy. Several research works have aimed at comparing the surface modification behavior and biocompatibility of Ti67 to the commercially used Ti64 [26,27]. Additionally, Sharma et al. [28] investigated Ti67 alloy in terms of surface integrity, bioactivity, and functional properties before and after EDM machining. It was found that surface machining using the WEDM process enhanced the bioactivity of the sites on the surface of Ti67 for the development of hydroxyapatite.

Based on the literature above, further research is needed to deeply understand the effect of WDEM process parameters on the surface quality of Ti67 alloy as compared to that of Ti64. In particular, the machining mode (number of cuts), exists as a process parameter in WEDM but is not taken into account during the operation process, except for a paper that discussed this factor and its effect on surface roughness [28]. The aim of the current study was, therefore, to do thorough research concerning the machining of the two Ti-alloys (Ti64 & Ti67) using WEDM. The main performance indexes of the WEDM process were determined under different cutting speeds and modes. Finally, mathematical models to predict the roughness of the machined surfaces based on the process parameter were employed and evaluated.

## 2. Experimental Procedures

The experimental work is summarized in the following flowchart of Figure 2. At first, two samples of Ti64 and Ti67 were prepared by casting. A Series of experiments were then conducted using WDEM. Surface roughness and micro-hardness were measured. Average recast layer thickness (ARLT) was determined using SEM micrographs of the sub-machined surface layer. Finally, based on Response Surface Methodology (RSM), mathematical equations representing the relationship between the surface roughness and the machining parameters were proposed and then evaluated using Analysis of Variance (ANOVA).

### 2.1. Casting

A photograph of the melting chamber of an arc furnace that works under a vacuum is shown in Figure 3. The furnace has three main units: a melting chamber, a control unit, and a chiller. As the first step, pieces of pure metals are placed in the water-cooled crucible, and the melting chamber is evacuated. When the vacuum reaches the required degree, argon gas is injected into the vacuum chamber then the high voltage is applied between the copper crucible and the arc gun. Once the arc touches the copper crucible, the arc is initiated, and the samples melt. Afterward, the vacuum is released, and the solidified samples are taken out. Table 1 shows the chemical composition of the obtained samples using the optical emission spectrometer (OES) model (OXFORD INSTRUMENTS).

### 2.2. WEDM Parameters and Sample Preparation

The experimental setup of CNC (WEDM) (Model: RIJUN-FH 300) and the position of the workpiece (Ti64/Ti67) is shown in Figure 4. A molybdenum wire (Mo W) was used as an electrode material. For the current experiment, the process parameters with their three levels are shown in Table 2. A total of 13 experiments were performed at different combinations of input parameters according to the run order (using Design of Experiments).

Changing the cutting mode was chosen in the current work due to its significant effect on the machined surfaces during WEDM. The rough-cut mode (RC) is done to remove the maximum amount of material. In contrast, the trim-cut mode uses three cuts (TRC-III) and seven trim cuts (TRC-VII) to decrease the recast layer and the surface defects to the minimum by removing only thin layer of the workpiece surface. The rim-cutting operation, surface machining, and other terminology are described in Figure 5.

After machining, the cross-section of the workpieces was metallographically prepared. The specimens were cut perpendicular to the machined surfaces and polished with 400 down to 2000 grit paper, followed by chemical etching in a solution of 15 mL HNO_3_, 10 mL H_2_SO_4_, 50 mL HF and 300 mL of water for 10 s. Finally, all the specimens were cleaned in an ultrasonic vibrator, and dried before analyzing the surface characteristics on a scanning electron microscope (SEM) to measure the recast layers (RL) thickness.

Surface quality was determined in terms of surface roughness (Ra) which was measured using a high-quality Mitutoyo Surftest SJ-201P instrument by taking five readings along the machined surface in different locations. Another parameter, micro-hardness underneath the WEDM surface was determined by a microhardness tester model (FUTURE TECH FM -800e), by calculating the micro-indents on the transverse polished surface using 200 gm load, dwell time 10 s and the obtained results were the average of five readings.

### 2.3. Response Characteristics

In the current investigation, surface characteristics of Ti-6Al-7Nb and Ti-6Al-4V alloy specimens were assessed as the response variables. To analyze the surface response, Response Surface Methodology (RSM), which combines both statistical and mathematical techniques, was used. In RSM, the correlation between response variables and explanatory variables are specified.

The average surface roughness values at different machining conditions (13 experiments) were used as inputs to RSM and mathematical equations were then produced for the two alloys to represent the form of relationship between response and independent variables [29]. The RSM can be mathematically expressed as [18,22].
(1)$$F\;(x_{1},\;x_{2},\;\cdots ,\;x_{k})\;=\;a_{0}+\underset{\mathrm{Linear}\;\mathrm{terms}}{\underbrace{\sum_{i=1}^{k}a_{i}x_{i}}}+\sum_{i=1}^{k}\;\;\underset{\mathrm{Squared}\;\mathrm{terms}}{\underbrace{a_{ii}\;{x_{i}}^{2}}}+\underset{\mathrm{Cross}-\mathrm{product}\;\mathrm{terms}}{\underbrace{\sum^{\;}\sum_{i\leq j}\;a_{ij}x_{i}}}\;$$

As a final step, ANOVA was used to define the competency of the RSM equation/model and the statistical significance of its terms. Actually, no rule exists concerning the order of model that fits to each problem. It is a trial-and-error process depending on the obtained results.

## 3. Results and Discussion

### 3.1. Original Microstructure of the Investigation Samples

Microstructure of the investigated samples, shown in Figure 6, is characterized by the well-known cast microstructure of two-phase titanium alloys that consists of Widmanstätten colonies of α with prior β grains. It is observed here that, Ti67 alloy has finer grain structure compared to Ti64. This difference in the grain size is expected to affect the surface response to machinability. Another essential difference between Ti64 and Ti67 is that Nb has been reported to resist oxidation in titanium alloys and hence minimizes the chance of alpha case formation which is the brittle layer that forms beneath the surface of the two-phase Ti-alloys upon oxidation [30,31]. Since the two alloys are different in their microstructures (in terms of grain size) and in their composition (different β stabilizing element), obvious differences in their machined surfaces are expected.

### 3.2. Surface Integrity of WEDM Samples

Figure 7a–f presents surfaces of Ti-6Al-4V and Ti-6Al-7Nb samples machined under different cutting modes. As noted from the pictures, the machined surfaces of alloys featured different colors and surface roughness after cutting conditions from Figure 7a to Figure 7e for Ti-6Al-4V and Figure 7b to Figure 7f for Ti-6Al-7Nb. This is most probably due to the difference in the chemical composition of the two alloys.

The SEM micrographs of the cross section of the machined Ti64 and Ti67 specimens under different cutting modes are shown in Figure 8a–f. The measurements of RL are also summarized in Figure 9. At rough cut (RC) the re-solidified or recast layer (RL) showed waving structure of variable thickness, which differs for the two alloys. In the case of Ti64, RL recorded the largest value around 30 ± 0.93 µm, Figure 9. This number declined sharply to half for Ti67. After the melted material is re-solidified, tensile stress distribution varies between the parent material and the top machined layer, leading to the generation of micro-cracks on the machined surface. The machined surface contains both overlapped craters that are resulted from the bombardment of ions induced due to the repetitive sparking and collapsing of vapor bubbles which causes melting and hence evaporation of material. This spark is a large space (refer to Figure 5) and depends on the wire offset value, thus influencing the thickness of the work material. Similarly, the machined surface after trim cutting operations, i.e., TRC-III and TRC-VII, varied in thickness, which can be observed in Figure 8. The RL witnessed a significant decrease from 17 ± 0.73 µm to 8 ± 0.23 µm for Ti64 and Ti67, respectively. In TRC-VII, the RL showed a continuous decline to 12 ± 1.31 µm for Ti64 and 5 ± 0.72 µm for Ti67, which was the lowest thickness of the recast layer, and shows better surface topography with low surface defects having a reduced thickness of RL and low Ra. In the trim cut, the spark zone was smaller than the rough cut (refer Figure 5), thus, removing a very thin layer of the material with low discharge energy.

### 3.3. Micro-Hardness Analysis

Micro-hardness of the machined specimens was measured underneath the machined surface, where micro-hardness represents the degree of thermal damage on the surface caused by WEDM. Figure 10a,b shows the micro-hardness of Ti64 and Ti67, respectively, at different machining modes under a speed of 100 µm/s. It can be observed that TRC-VII machined specimens had higher values of micro-hardness at 20 µm below the machined surface (390 ± 30 HV), and it decreased towards the underneath of the machined surface and obtained the value of the original material (315 ± 50 HV) for Ti64. Alloy Ti67, followed the same trend but with (480 ± 55 HV) at TRC-VII, which was higher than that of Ti64.

It has been reported that the heat generated in the machining zone is partly used in melting and vaporizing the workpiece material. A part of this heat is conducted to the workpiece, which is very small due to the lower thermal conductivity and the rest is absorbed by the dielectric. Therefore, temperature of the machined surface is considerably greater than that bottom layers. This causes intense heat concentration in the thin layer of machined surface and a significant temperature gradient is experienced along the depth of the workpiece.

The possible reason for the increased micro-hardness below the top layer of the machined surface is the metallurgical transformation because of the quenching process due to the continuous circulation of the dielectric. Therefore, the properties of the machined surface change significantly compared to that of bulk material. Additionally, the resistance to plastic deformation of the trim-cut surface was higher than that of the rough-cut surface due to the relatively lower number of micro-pores and cracks resulting in its higher micro-hardness.

From Figure 11, it can be seen that the recast layers and heat-affected zone were prominent on machined titanium alloy surface due to the high temperature on the surface and significant temperature gradient towards the bulk material. The machined surfaces of Ti64 contained thick recast layer, observed holes near the subsurface, and cracks were pervasive on the workpiece. Thin recast layers and discontinuous flaky type islands on the top of the solid layer, which did not contain any cracks, were detected in Ti67 as shown in the cross-section images.

### 3.4. Analysis of Surface Roughness (SR)

The surface roughness of Ti64 and Ti67 alloys under different machining modes and cutting speeds (50, 100 µm/s) are analytically presented in Figure 12a,b. The average surface roughness for the WEDM surface of samples at cutting speed 50 µm/s varied from RC to TRC-VII. At rough cut, a significant difference was observed in the surface roughness of Ti64 and Ti67 alloys which recorded 5.68 ± 0.44 and 4.52 ± 0.35 µm, respectively. This was followed by sharp decrease in roughness to 1.37 ± 0.12 µm for Ti64 and 1.62 ± 0.18 µm for Ti67 at TRC-III. Finally, at TRC-VII, surface roughness declined to 1.02 ± 0.20 µm for Ti64 compared to 0.92 ± 0.10 µm for Ti67, which considered the lowest value at those conditions represented in Figure 10. At a cutting speed of 100 µm/sec, surface roughness at rough cut for both alloys showed the highest level. By increasing number of cuts to three cuts (TRC-III), surface roughness for Ti64 and Ti67 significantly increased and reached to 2.34 ± 0.28 µm and 1.42 ± 0.15 µm, respectively. Ti-6Al-4V alloy achieved a slightly lower surface roughness of 0.88 ± 0.12 µm compared to 0.9 ±0.08 µm for Ti-6Al-7N at TRC-VII. Generally, variation in surface properties due to cutting speed and the number of cuts is due to the heat energy effect across the electrodes, which leads to melting and erosion of the working material [28]. In rough cutting mode, higher thermal damage on the machined surface is due to the higher melting and evaporation than the trim cutting operation.

Surface roughness values obtained in Figure 12 are summarized in Table 3 for the 13 experiments that were suggested by design of experiments. Based on these input data, mathematical Equations (2) and (3) were developed and used to analysis roughness of Ti64 (Ra_1_) and Ti67 (Ra_2_), respectively.
Ra_1_ = 14.34 − 0.130 A − 5.606 B + 0.0386 AB + 0.0005 A² + 0.582 B ² − 0.0044 AB²(2)
Ra_2_ = 1.74 + 0.138 A − 1.9999 B − 0.0008 AB − 0.0009 A² + 0.169 B ² + 0.0003 AB²(3)
where A is the speed (µm/s) and B is the number of cuts.

For analyzing roughness output characteristics, the 3D response has been plotted in Figure 13, as a function of cutting speed and number of cuts for the two alloys.

It was found from the plot that, at low cutting speed, if the number of cuts is increased from 1 to 7, a small change is observed in Ra value. However, at the different number of cuts, it was apparent that changing the cutting speed had a low effect on SR and the change in the number of cuts had a significant effect on SR (Ra_1_). Although, it can be seen that, in Figure 9b for Ti67, at a low number of cuts, changing cutting speed created a tipping point of SR at the middle of cutting speed, while at low, moderate, and high number of cuts, changing the cutting speed had a low effect on the surface roughness (Ra_2_). Generally, the dominant parameter controlling the surface roughness for both alloys is the number of cuts followed by cutting speed. It is difficult to estimate the predicted values of new points from the 3D curve, therefore, it was decided to construct contour curves [30] as shown in Figure 14.

### 3.5. Analysis of Variance (ANOVA) Analysis

Figure 15 and Figure 16 illustrate the proposed models for Ti64 and Ti67 based on ANOVA. The statistical summary reveals that the SR model is significant, and the lack of fit is nonsignificant; the appropriate fit is further supported by the model’s *p*-value, which is less than the 0.05 necessary for a satisfactory ANOVA. In the ANOVA, the sum of squares value determines the proportion effect of each component on SR. The number of cuts had the greatest influence on SR, followed by the quadratic term of the number of cuts, the interaction of cutting speed, quadratic term of the number of cuts, cutting speed, the interaction of cutting speed and number of cuts, and quadratic term of cutting speed. This conclusion was completely consistent with experimental results. R^2^ indicated that the current model can predict 99.57% of future outcomes by the present model due to significant and nonsignificant terms. Adj-R^2^ displays the percentage prediction (99.13%) due to significant terms only. For an acceptable indication of ANOVA, the difference between Adj-R^2^ and Pred-R^2^ should be less than 20%. Pred-R^2^ and Adj-R^2^ in this study had a close agreement and the value of adequate precision was 43.75. An adequate precision value greater than 4 is a sign of good ANOVA. The model’s theoretically assessed suitable fit is displayed in Figure 15 and Figure 16, where the Ra and Ra predicted values were plotted with only a slight variance.

The normal probability distribution curve is depicted in Figure 17. This figure clearly shows that all residuals were detected in a straight line. As a result, the residual distribution was normal. The regular distribution of predicted and residual values is also shown in Figure 18. Thus, all of the results were located on a straight line, indicating a satisfactory ANOVA. A deviation from the straight line shows the deterioration of the ANOVA quality.

## 4. Conclusions

In the present research, Ti-6Al-4V and Ti-6Al-7Nb alloys were machined by WEDM using two parameters cutting speed and machining modes, rough cut (RC) and trim cut (TRC-III and TRC-VII) of WEDM. In addition, Ra, RL and micro- hardness were calculated as the major machining performance indexes. For all the above-mentioned process indexes ANOVA was performed, while the Response Surface Methodology was employed in order to define the correlation between machining parameters and built a mathematical model. In brief, the deduced conclusions of the current study are:The cast microstructure of Ti67 showed finer grain size compared to Ti64;Rough cut mode WEDM produced high thermal damage on the workpiece surface, resulting in a high value of roughness (Ra-value) and thick recast layer. Contrasting trim cut modes TRC-III and TRC-VII led to enhanced surface properties for both alloys;The Ra of machined samples at cutting speed 50 µm/s under RC varied for Ti64 and Ti67 alloys were 5.68 ± 0.44 and 4.52 ± 0.35 µm, respectively. They also decreased to 1.37 ± 0.12 µm and 1.62 ± 0.18 µm at TRC-III. At seven number of cuts TRC-VII, it declined to 1.02 ± 0.20 µm for Ti64, compared to 0.92 ± 0.10 µm, for Ti67. Surface roughness followed the same behavior at 100 µm/s, in which, Ra values for Ti67 alloy declined from 4.2 ± 0.34 µm at RC to 0.96 ± 0.08 µm at TRC-VII. This value also dropped from 4.4 ± 0.45 at RC to 0.88 ± 0.12 µm at TRC-VII which was the smallest value in this case;TRC-VII significantly reduced the thickness of recast layer to 5 ± 0.72 µm from 12 ± 1.31 µm obtained in RC on machined surface for Ti67 alloy. For Ti64, the re-solidification layer 30 ± 0.93 µm at RC witnessed a significant decline to 12 ± 1.31 µm at rough cut;The measured micro-hardness was very dependent on cutting mode. At TRC-VII, TRC-II and RC were (390 ± 30 HV, 385 ± 35 HV and 315 ± 15 HV) for Ti64 and (480 ± 55 HV, 460 ± 50 HV, and 320 ± 45 HV) for Ti67, respectively;It was observed that, in WEDM, the most influential parameters on surface roughness were the machining mode, followed by cutting speed and the interaction between them, where Response Surface Methodology (RSM) defined the correlation between machining parameters and built a mathematical model. ANOVA table proved the strength of the model equation and the reliability of obtained data;Replacing elemental V with Nb led to enhancing the machinability of titanium alloy as a function of surface roughness, recast layer, and microhardness.

## Figures and Tables

**Figure 1 materials-16-00688-f001:**
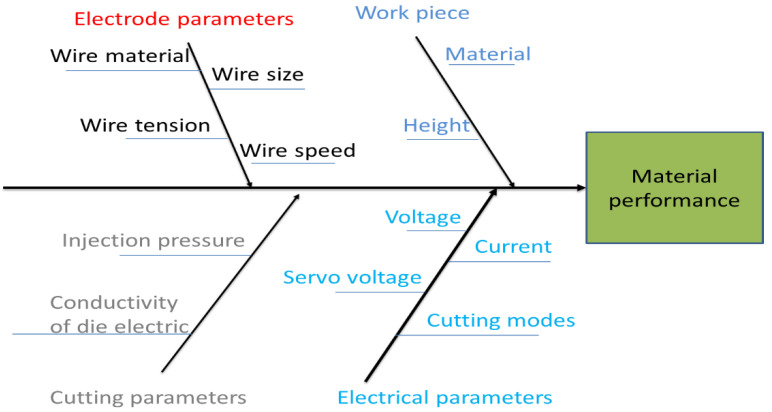
Ishikawa Cause and Effect Diagram for WEDM Process [15].

**Figure 2 materials-16-00688-f002:**
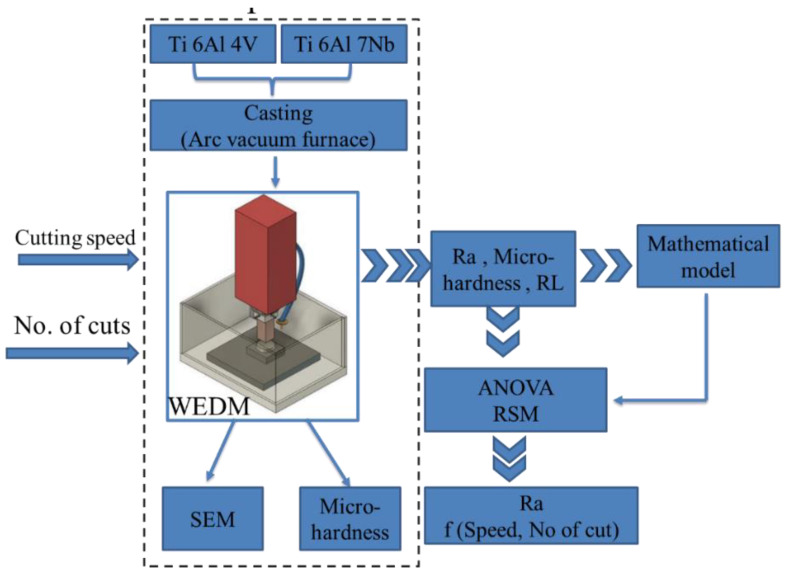
Experimental procedures flow chart.

**Figure 3 materials-16-00688-f003:**
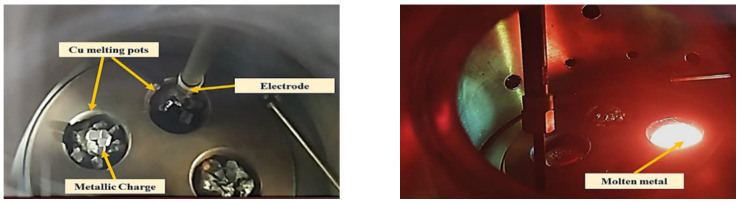
Placing the charge into the melting pot in the vacuum arc furnace.

**Figure 4 materials-16-00688-f004:**
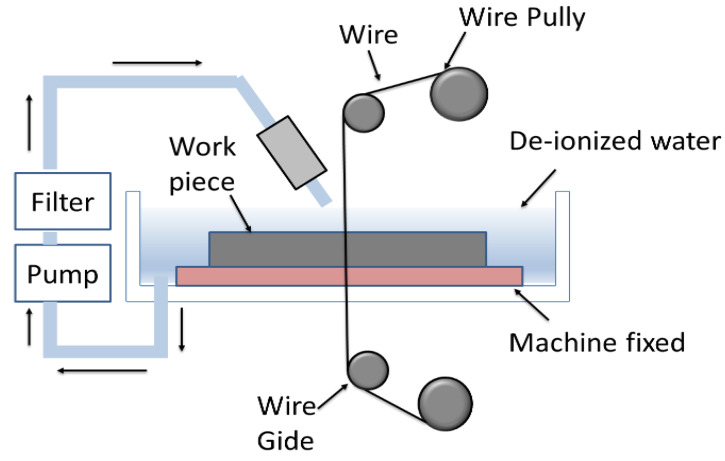
Wire electric discharge machining (WEDM) set up.

**Figure 5 materials-16-00688-f005:**
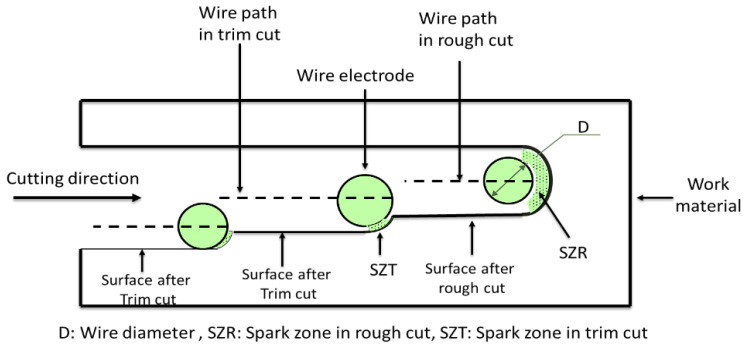
The principal of rough and trim cut modes of WEDM, reproduced, with permission from Springer (1293523) [18].

**Figure 6 materials-16-00688-f006:**
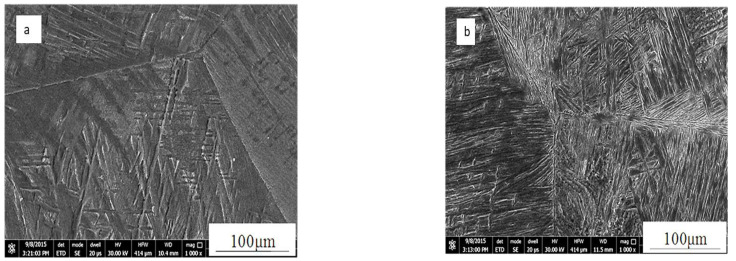
Microstructure of cast samples: (**a**) Ti-6Al-4V, and (**b**) Ti-6Al-7Nb.

**Figure 7 materials-16-00688-f007:**
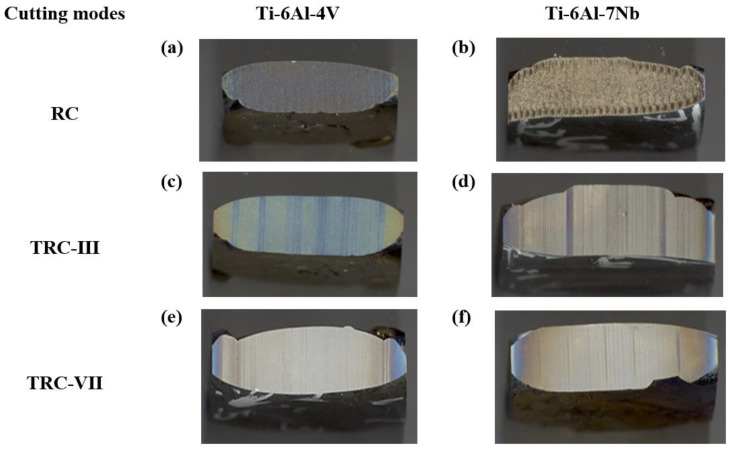
Images of machined surfaces at different cutting conditions.

**Figure 8 materials-16-00688-f008:**
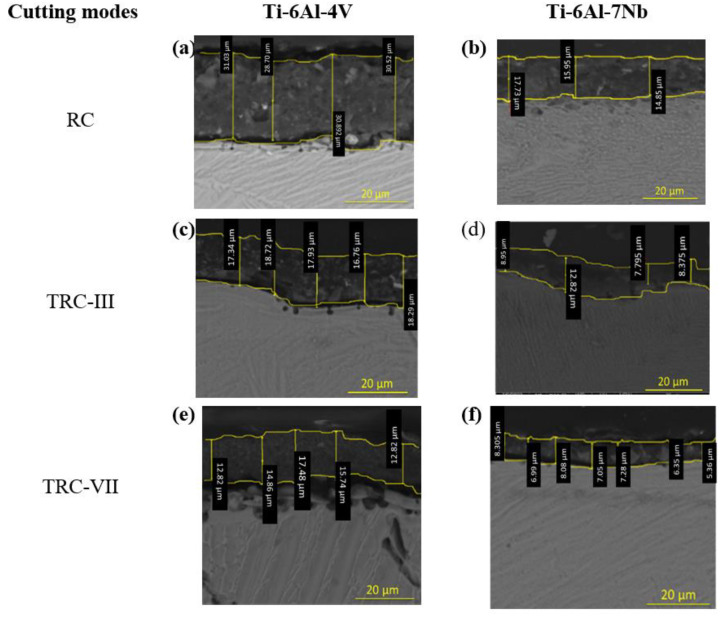
SEM micrographs under different cutting modes that show the transverse surface along with the thickness measurements of recast layer.

**Figure 9 materials-16-00688-f009:**
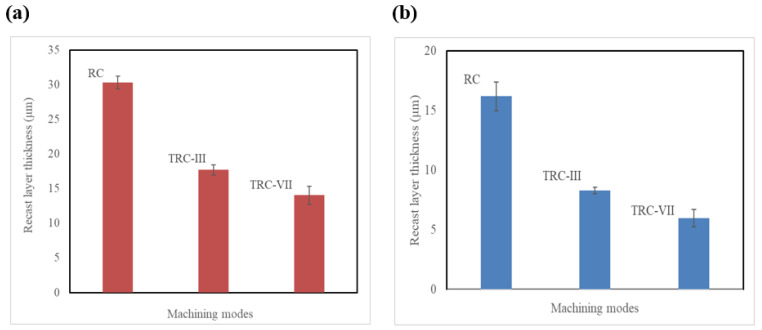
Results of recast layer thickness measurements of machined specimens (**a**) Ti-6Al-4V and (**b**) Ti-6Al-7Nb.

**Figure 10 materials-16-00688-f010:**
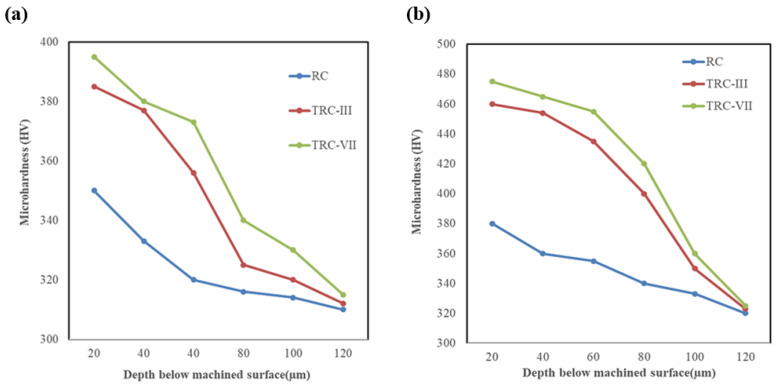
Distribution of micro-hardness underneath the machined surface, (**a**) Ti-6Al-4V and (**b**) Ti-6Al-7Nb.

**Figure 11 materials-16-00688-f011:**
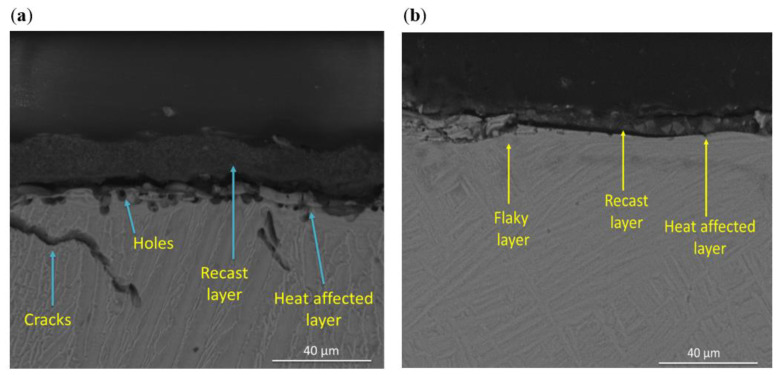
Typical cross-sectional view of surface machined by wire EDM: (**a**) Ti-6Al-4V and (**b**) Ti-6Al-7Nb.

**Figure 12 materials-16-00688-f012:**
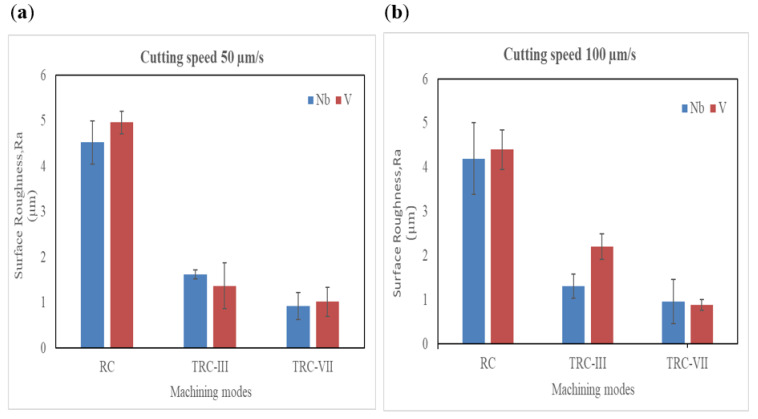
Surface roughness of Ti-6Al-4V and Ti-6Al-7Nb: (**a**) Comparison at cutting speed 50 µm/s, (**b**) at 100 µm/s of machined samples.

**Figure 13 materials-16-00688-f013:**
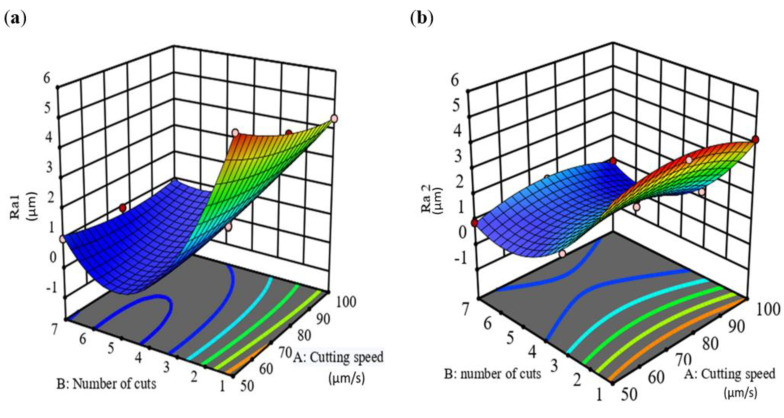
Effect of cutting process variables on surface roughness 3D for (**a**) Ti-6Al-4V (Ra_1_) and (**b**) Ti-6Al-7V (Ra_2_) alloys.

**Figure 14 materials-16-00688-f014:**
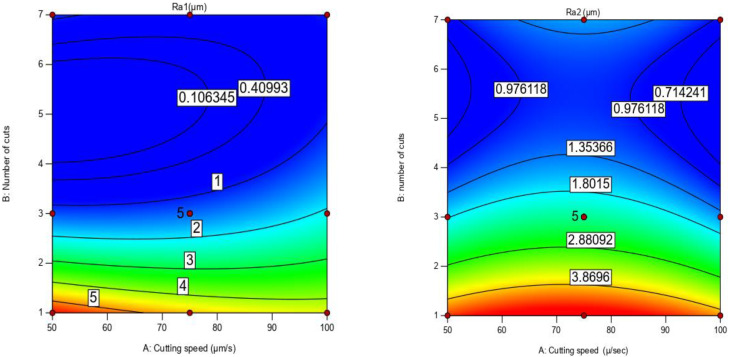
Effect of cutting process variables on surface roughness contour for Ti-6Al-4V (Ra_1_) and Ti-6Al-7Nb (Ra_2_) alloy.

**Figure 15 materials-16-00688-f015:**
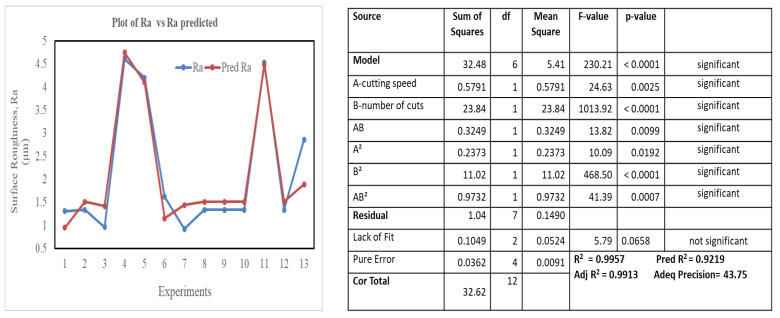
Plot of Ra vs. predicted Ra, ANOVA model for Ti-6Al-4V.

**Figure 16 materials-16-00688-f016:**
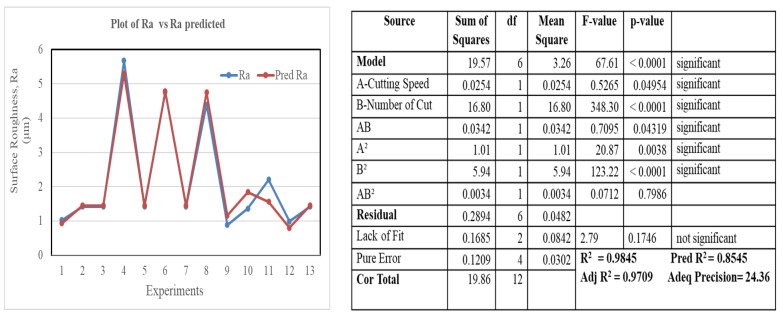
Plot of Ra vs. predicted Ra, ANOVA model for Ti-6Al-7Nb.

**Figure 17 materials-16-00688-f017:**
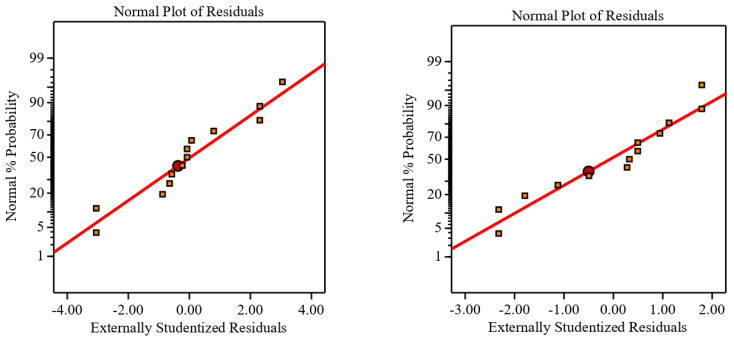
Normal probability plot in the left for Ti-6Al-4V and in the right for Ti-6Al-7Nb.

**Figure 18 materials-16-00688-f018:**
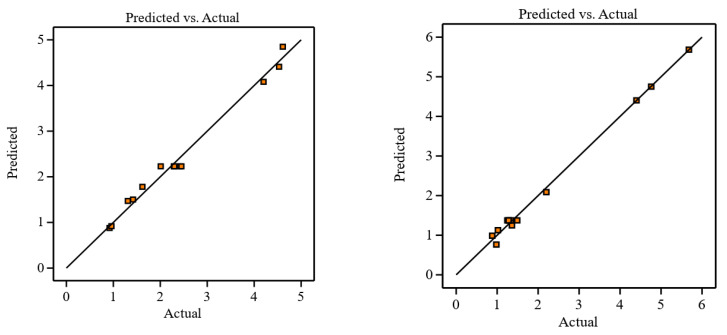
Predicted versus actual plot of residuals in the left for Ti-6Al-4V and in the right for Ti-6Al-7Nb.

**Table 1 materials-16-00688-t001:** Chemical composition of titanium alloys.

**Ti-6Al-4V**	**Element**	**Al**	**V**	**Fe**	**C**	**O**	**N**	**Others**		**Ti**
	wt. %	6.2	4.0	0.1	0.02	0.03	0.01	≤0.4		Balance
**Ti-6Al-7Nb**	**Element**	**Al**	**Nb**	**Fe**	**C**	**O**	**N**	**Others**	**Ta**	**Ti**
	wt. %	6.2	6.8	0.03	<0.01	0.14	<0.01	≤0.4	<0.05	Balance

**Table 2 materials-16-00688-t002:** Process parameters and their levels.

Parameter (Level)	Cutting Speed µm/s	Number of Cuts
1	50	1
2	75	3
3	100	7
Factor	A	B

**Table 3 materials-16-00688-t003:** Experimental plan and related readings.

			Ti-6Al-4V	Ti-6Al-7Nb
Exp No	Cutting Speed µm/s	Number of Cuts	Surface Roughness (μm)	
1	50	1	5.68	4.53
2	100	1	4.40	4.20
3	50	7	1.02	0.92
4	100	7	0.88	0.96
5	50	3	1.36	1.62
6	100	3	2.20	1.31
7	75	1	4.76	4.61
8	75	7	0.98	2.86
9	75	3	1.49	1.34
10	75	3	1.29	1.30
11	75	3	1.25	1.33
12	75	3	1.34	1.35
13	75	3	1.28	1.31

## Data Availability

All the available data is provided in the manuscript.

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
