# Peer review of "Influence of Microstructure and Alloy Composition on the Machinability of α/β Titanium Alloys"

_materials, 2023, doi:10.3390/ma16020688_

Round 1

Reviewer 1 Report

The manuscript describes a comparative study of the machinability of two α/β titanium alloys, namely Ti-6Al-4V (Ti64) and Ti-6Al-7Nb 29 (Ti67), analyzing the effects of two different input parameters used during WEDM: cutting speed and cutting mode, on the surface responses.

The novelty is not apparent because machinability tests are commonly done, so a novelty and significance statement is needed upfront. The method is appropriate for achieving the objectives, but there lack experimental details, i.e., the surface roughness, hardness, SEMs analyses’ applied operating parameters, and so on, lowering the reproducibility of the study. There also lacks statistical analysis of these determinations, the sample number n of each experimental group has not been given, and the results are presented with absolute values without standard deviations, raising the question about the statistical significance.

Hence, a major revision is needed.

Some details for the authors’ consideration:

1. The Abstract is overly long and repetitive, please refine.

2. The references are outdated. Here it is recommended to also cite the most up-to-date literature where the effects of surface properties and chemical composition on cell/tissue adhesion and other performances discussed for biomedical applications have been thoroughly [5, 10.3390/met12030406]

3. Figures, captions all need to be re-written to describe the figures, simple “Graphical overview.” is unacceptable. For multi-panel figures (3, 4), the content of each panel needs to be described. 4 change arrow color to white to increase visibility. 5, is this complete reuse of this Fig. with permission? 6, 9 and Table 5, must state what kind of images, magnifications used and bar scales.

4. Plots, axises lack ticks.

5. Please keep the same order, e.g. M&M, roughness then hardness, results should be presented in the same order. 64 then 67, Fig. 10 content should be in same order.

6. Which instrument was used to measure hardness?.

7. Source of Table 1 should be given.

8. Table 3 is not necessary, can simply write them out in text.

9. Are these 3 cutting modes routine operations? If so, can simply state the effect of each was examined, rather than “Changing the cutting mode was chosen..”

10. Avoid using subjective terms like “high-quality” “obvious”

11. It is not clear how thickness was determined

12. Which program was used for the ANOVA analysis?

13. Discussion and conclusion do not reflect on the topic of the study, machinability. What is the conclusion on machinability? Which alloy has higher machinability? And the underlying mechanism?

Author Response

Reviewer 1

Comments and Suggestions for Authors

The manuscript describes a comparative study of the machinability of two α/β titanium alloys, namely Ti-6Al-4V (Ti64) and Ti-6Al-7Nb 29 (Ti67), analyzing the effects of two different input parameters used during WEDM: cutting speed and cutting mode, on the surface responses.

The novelty is not apparent because machinability tests are commonly done, so a novelty and significance statement is needed upfront. The method is appropriate for achieving the objectives, but there lack experimental details, i.e., the surface roughness, hardness, SEMs analyses’ applied operating parameters, and so on, lowering the reproducibility of the study. There also lacks statistical analysis of these determinations, the sample number n of each experimental group has not been given, and the results are presented with absolute values without standard deviations, raising the question about the statistical significance. Hence, a major revision is needed.

Reply: the weak points in the experimental part were modified with more details as recommended.

Some details for the authors’ consideration:

  1. The Abstract is overly long and repetitive, please refine.

Reply: We tried to decrease the abstract a little; however, further decreasing will hide some important details that are essential to the manuscript.

  1. The references are outdated. Here it is recommended to also cite the most up-to-date literature where the effects of surface properties and chemical composition on cell/tissue adhesion and other performances discussed for biomedical applications have been thoroughly [5, 10.3390/met12030406]

      References added

  1. Figures, captions all need to be re-written to describe the figures, simple “Graphical overview.” is unacceptable. For multi-panel figures (3, 4), the content of each panel needs to be described. 4 change arrow color to white to increase visibility. 5, is this complete reuse of this Fig. with permission? 6, 9 and Table 5, must state what kind of images, magnifications used and bar scales.

 Reply: modifies as required

4.Plots, axises lack ticks.

       Modified as recommended

  1. Please keep the same order, e.g. M&M, roughness then hardness, results should be presented in the same order. 64 then 67, Fig. 10 content should be in same order.

       Modified as recommended

  1. Which instrument was used to measure hardness?.

       Added to the experimental section

  1. Source of Table 1 should be given.

       Added to experimental section

  1. Table 3 is not necessary, can simply write them out in text.

       Deleted as recommended

  1. Are these 3 cutting modes routine operations? If so, can simply state the effect of each was examined, rather than “Changing the cutting mode was chosen..”

        Reply: This option is in the machine, but it is not used. Investigating the results of applying it  is the novelty of our research.

  1. Avoid using subjective terms like “high-quality” “obvious”

Done

  1. It is not clear how thickness was determined

Reply: Some details have been added to the paper to clear how recast layer was measured.  

  1. Which program was used for the ANOVA analysis?

Reply: Design expert 13

  1. Discussion and conclusion do not reflect on the topic of the study, machinability. What is the conclusion on machinability? Which alloy has higher machinability? And the underlying mechanism?

Reply: By replacing V with Nb element leads to enhance the machinability of titanium alloy as a function of surface roughness, recast layer, and micro hardness

Thank you for your kind recommendations

Reviewer 2 Report

The paper must be revised to facilitate readership and a better scientific notation, particularly regarding the following issues:

(1) Units must be always written in the same way (for instance, seconds are sometimes called "sec" and other times "s"; both “µ” and “µm” are employed). Do not use names that are not scientific such as "microns". Use HV instead of Hv.

(2) There is no need to use alternate acronyms for referring to the alloys, but always use "Ti-6Al-4V" and "Ti-6Al-7Nb" in the text and in the legends of figures and tables. 

Additionally notice that Ti-6Al-4Nb in line 155 is wrong.

(3) Tables 4 and 5 are actually better described as figures, and labels (a)-(d) should be added for reference in the body of the manuscript when referring to one of the figures, instead of referring in general to the complete Table/Figure.

In addition, scale bars must be included in both cases.

(4) Labels (a) and (b) should also be employed in Figure 10, and explain in the legend.

(5) Use units in the axes of all graphs. They are missing in Figures 11 and 12.

(6) SEM images should only contain the scan bar as information, removing the machine model, etc. Use the deleted space to include the (a) and (b) levels instead of placing a box on the micrographs.

(7) References are not organized using the rules for the Journal

Other corrections needed:

line 43 - The micro-hardness values correspond to the surface of the machined layer, and they vary with depth.

line 159 - The sentence "a seul research for Sharma" is wrong

line 162 - Revise "enhance the bioactivity sites". Either is referring to the number of bioactive sites, or the bioactivity of the sites.

line 286 - The surface preparation described is not "polishing" but "abrasion".

line 288 - Give the solvent used in the sonication cleaning.

line 307 - The terms in the equation are wrongly assigned, and "j" must be underscript in the last term at the right

In addition, describe the meaning of the symbols in the body of the text.

line 394 - Give the depth at which the bulk property is attained.

lines 410-415 - Start the paragraph calling to the figure that is described here.

Figure 8 (a) - Is there a new sample TRC-I?

Figures 13 and 14 - In the X axis, "experiments"

Author Response

Reviewer 2

Comments and Suggestions for Authors

The paper must be revised to facilitate readership and a better scientific notation, particularly regarding the following issues:

(1) Units must be always written in the same way (for instance, seconds are sometimes called "sec" and other times "s"; both “µ” and “µm” are employed). Do not use names that are not scientific such as "microns". Use HV instead of Hv.

  • Modified as recommended

(2) There is no need to use alternate acronyms for referring to the alloys, but always use "Ti-6Al-4V" and "Ti-6Al-7Nb" in the text and in the legends of figures and tables. 

  • (done)

Additionally notice that Ti-6Al-4Nb in line 155 is wrong.

  • ( corrected)

(3) Tables 4 and 5 are actually better described as figures, and labels (a)-(d) should be added for reference in the body of the manuscript when referring to one of the figures, instead of referring in general to the complete Table/Figure.

In addition, scale bars must be included in both cases.

  • (added)

(4) Labels (a) and (b) should also be employed in Figure 10, and explain in the legend.

  • (done)

(5) Use units in the axes of all graphs. They are missing in Figures 11 and 12.

  • (done)

(6) SEM images should only contain the scan bar as information, removing the machine model, etc. Use the deleted space to include the (a) and (b) levels instead of placing a box on the micrographs.

  • (done)

(7) References are not organized using the rules for the Journal Other corrections needed:

line 43 - The micro-hardness values correspond to the surface of the machined layer, and they vary with depth.

line 159 - The sentence "a seul research for Sharma" is wrong

  • (done)

line 162 - Revise "enhance the bioactivity sites". Either is referring to the number of bioactive sites, or the bioactivity of the sites.

  • (the bioactivity of the sites)

line 286 - The surface preparation described is not "polishing" but "abrasion".

  • This is the sequence of preparation workpiece (polishing then applied etching)

line 288 - Give the solvent used in the sonication cleaning.

  • (Alcohol)

line 307 - The terms in the equation are wrongly assigned, and "j" must be underscript in the last term at the right

  • (corrected)

In addition, describe the meaning of the symbols in the body of the text.

  • (done)

line 394 - Give the depth at which the bulk property is attained. ( from the figure 10 refered the started Depth, where micro hardeness was meachered

lines 410-415 - Start the paragraph calling to the figure that is described here.

  • (done)

Figure 8 (a) - Is there a new sample TRC-I?

  • No, its modified in new version

Figures 13 and 14 - In the X axis, "experiments"

  • (Done)

Thank you for your kind recommendations

Round 2

Reviewer 1 Report

Authors tried to address all the issues raised, and the overall quality and reproducibility have been improved.

The track changes and with figures overlaying make the reviewing process difficult. Please return with a clean version with all changes accepted, highlighing changes for proper reviewing. 

It still is not clear if Fig. 1, 5 are complete reuse of the figs from publications with permission or are reproduced/inspried by publications? Must specify.

SEM micrographs, caption must specify magnification applied, scale bar.

Change marking color in Fig. 8 from red to white or yellow.

The original comment "There also lacks statistical analysis of these determinations, the sample number n of each experimental group has not been given, and the results are presented with absolute values without standard deviations, raising the question about the statistical significance." still applies.

18 figures are too many, distracting readers from the essential science. Suggestion: shrink some and combine to make multi-panel figures. Some less important ones (e.g. Avova) can go into supplementary information.

Author Response

Dear Editors and Reviewers

Thank you for your efforts in revising our submitted work. The reviewers’ recommendations were considered as follows:

Reviewer 1

-The track changes and with figures overlaying make the reviewing process difficult. Please return with a clean version with all changes accepted, highlighting changes for proper reviewing. 

Reply: the required changes have been highlighted in red in the manuscript.

-It still is not clear if Fig. 1, 5 are complete reuse of the figs from publications with permission or are reproduced/inspried by publications? Must specify.

Reply: - Fig 1, inspired by publications (The paper content is free to use under Creative common license)

              Fig 5, reproduced, with permission from Springer (1293523)

-SEM micrographs, the caption must specify the magnification applied and the scale bar. Change the marking color in Fig. 8 from red to white or yellow.

Reply: - changed

 -The original comment "There also lacks statistical analysis of these determinations, the sample number n of each experimental group has not been given, and the results are presented with absolute values without standard deviations, raising the question about the statistical significance." still applies.

  Reply:  two samples were divided into 13 slides (number of experiments as shown in table 3).  Standard deviations were added in text and graphs.  Analysis Of Variance (ANOVA) is considered a statistical tool.

- figures are too many, distracting readers from the essential science. Suggestion: shrink some and combine to make multi-panel figures. Some less important ones (e.g. Anova) can go into supplementary information.

Reply: Analysis of Variance (ANOVA) is an important part of the paper.

Reviewer 2 Report

The authors have tried to improve the manuscript according to the reviewer's comments, although not achieved in all the occurrences as indicated below,

line 33 - "µm/s" instead of "µ/sec". The same in lines 36, 401, 435, inside of table 3 (line 462, cutting speed), inside Figure 13 (A axes in right and left), line 558

Replace "µ" by "µm" in lines 34 (2 times), 36 (2 times), 37, 38 (3 times), 439 (2 times), 440 (2 times), 444 (2 times), 445 (2 times), Y axes in Figure 12 (2 times), inside of table 3 (line 462, surface roughness), inside Figure 13 (Y axes in right and left), line 557 (3 times), line 558 (2 times), line 559 (2 times), line 560

Replace "Hv" by "HV" (still one missing occurrence in line 44 

Line 79 - The acronym WEDM is used in the body of the manuscript without explanation, well ahead of explaining EDM in line 105 (that is unneccessary)

Line 305 - Replace "sec" by "s"

Equation in line 319 - Edit as indicated in the image

Line 483 - Correct "AVOVA"

Units missing in A axes in Figure 14, left ande right

Units missing in Y axis (Roughness) in Figure 16 a-c

Line 561-563 - Use "µm" instead of "µ m"

Author Response

Dear Editors and Reviewers

Thank you for your efforts in revising our submitted work. The reviewers’ recommendations were considered as follows:

Reviewer 2

-line 33 - "µm/s" instead of "µ/sec". The same in lines 36, 401, 435, inside of table 3 (line 462, cutting speed), inside Figure 13 (A axes in right and left), line 558

 done

-Replace "µ" by "µm" in lines 34 (2 times), 36 (2 times), 37, 38 (3 times), 439 (2 times), 440 (2 times), 444 (2 times), 445 (2 times), Y axes in Figure 12 (2 times), inside of table 3 (line 462, surface roughness), inside Figure 13 (Y axes in right and left), line 557 (3 times), line 558 (2 times), line 559 (2 times), line 560

Reply: Surface roughness (Ra) measured unit (µm), while Recast layer (RL) measured unit (µm), Cutting speed (µm/s)

-Replace "Hv" by "HV" (still one missing occurrence in line 44 (done)

-Line 79 - The acronym WEDM is used in the body of the manuscript without explanation, well ahead of explaining EDM in line 105 (that is unneccessary)

Reply: WEDM has been explained in detail in the introduction section. The authors agree with the reviewer that EDM is unnecessary.

Line 305 - Replace "sec" by "s" (done)

Equation in line 319 - Edit as indicated in the image

Line 483 - Correct "AVOVA" (done)

Units missing in A axes in Figure 14, left and right (this is number, there is no units)

Units missing in Y axis (Roughness) in Figure 16 a-c

Line 561-563 - Use "µm" instead of "µ m" (done)
